# Rapid, Refined, and Robust Method for Expression, Purification, and Characterization of Recombinant Human Amyloid beta 1-42

**DOI:** 10.3390/mps2020048

**Published:** 2019-06-07

**Authors:** Priya Prakash, Travis C. Lantz, Krupal P. Jethava, Gaurav Chopra

**Affiliations:** 1Department of Chemistry, Purdue University, West Lafayette, IN 47907, USA; prakashp@purdue.edu (P.P.); tclantz@purdue.edu (T.C.L.); kjethava@purdue.edu (K.P.J.); 2Purdue Institute for Drug Discovery, Purdue University, West Lafayette, IN 47907, USA; 3Purdue Institute for Integrative Neuroscience, Purdue University, West Lafayette, IN 47907, USA; 4Purdue Institute for Inflammation, Immunology and Infectious Disease, Purdue University, West Lafayette, IN 47907, USA; 5Purdue Center for Cancer Research, Purdue University, West Lafayette, IN 47907, USA

**Keywords:** amyloid beta, recombinant abeta, HPLC, expression, purification, neuroscience, peptide, neurodegeneration, Alzheimer’s disease

## Abstract

Amyloid plaques found in the brains of Alzheimer’s disease patients primarily consists of amyloid beta 1-42 (Aβ42). Commercially, Aβ42 is synthesized using high-throughput peptide synthesizers resulting in the presence of impurities and the racemization of amino acids that affects its aggregation properties. Furthermore, the repeated purchase of even a small quantity (~1 mg) of commercial Aβ42 can be expensive for academic researchers. Here, we describe a detailed methodology for robust expression of recombinant human Aβ(M1-42) in Rosetta(DE3)pLysS and BL21(DE3)pLysS competent *E. coli* using standard molecular biology techniques with refined and rapid one-step analytical purification techniques. The peptide is isolated and purified from transformed cells using an optimized reverse-phase high-performance liquid chromatography (HPLC) protocol with commonly available C18 columns, yielding high amounts of peptide (~15–20 mg per 1 L culture) within a short period of time. The recombinant human Aβ(M1-42) forms characteristic aggregates similar to synthetic Aβ42 aggregates as verified by western blotting and atomic force microscopy to warrant future biological use. Our rapid, refined, and robust technique produces pure recombinant human Aβ(M1-42) that may be used to synthesize chemical probes and in several downstream in vitro and in vivo assays to facilitate Alzheimer’s disease research.

## 1. Introduction

Amyloid beta 1-42 (Aβ42) is a small ~4 kDa peptide produced when the amyloid precursor protein expressed on neuronal membranes is sequentially cleaved by β-secretase and γ-secretase [1]. Aβ exists as several variants ranging from 36 to 43 amino acid residues [2]. However, the main component of the toxic amyloid plaques found in the brains of Alzheimer’s Disease (AD) patients is composed of the Aβ42 isoform [3]. The extracellular accumulation of Aβ42 in the brain over time contributes to neuronal dysfunction and death leading to progressive memory loss and cognitive decline [4]. 

Several therapeutics currently in preclinical and clinical trials for AD are focused on targeting the cellular and molecular mechanisms related to Aβ42 [5,6]. Thus, there is an immediate need for further understanding of Aβ42’s biological function and its effect on both neurons and non-neuronal glial cells [7,8,9]. Researchers commonly use commercially-available synthetic Aβ42 for their experiments. However, the repeated purchase of synthetic Aβ42 can be expensive (~ $300 for 1 mg of peptide). Synthesizing Aβ42 in a traditional biochemistry laboratory has additional hurdles, such as 1) being expensive due to high instrument costs and 2) challenging due to the high hydrophobicity of the peptide that can affect the yield and efficiency of the procedure. The C-terminal sequence of Aβ42 in particular is known to be resistant to solid-phase peptide synthesis and is therefore called a “difficult sequence” peptide [10]. Furthermore, the presence of impurities and the racemization of amino acids during the synthesis of Aβ affects its aggregation properties. It has been shown that synthetic Aβ has lower aggregation kinetics and decreased neurotoxicity compared to recombinant human Aβ [11]. Thus, recombinant human Aβ is not only cheaper and easier to produce than the synthetic peptide but it is also free of enantiomers and is therefore better for investigating the biochemical and biological activity of Aβ.

Here, we present an alternate and refined approach for the rapid, easy, and low-cost production and purification of recombinant human Aβ42 containing an exogenous N-terminus methionine, denoted as Aβ(M1-42). The pET-Sac-Abeta(M1-42) plasmid was developed previously by Walsh et al. and expresses Aβ(M1-42) in *E. coli* cells [12]. In this original protocol, a combination of anion-exchange chromatography and centrifugal filtration was used to purify the peptide. Further, it was demonstrated that the exogenous methionine does not affect the kinetics and the fibrillation process of Aβ(M1-42) compared to Aβ42 [12]. Thus, the recombinant human Aβ(M1-42) and synthetic Aβ42 both exhibit aggregate-forming properties. The concentration-dependent aggregation kinetics of the Aβ(M1-42) and its variants with different N-terminal sequence extensions have also been previously evaluated [13]. In this protocol, we will focus on the expression of the recombinant Aβ(M1-42) peptide, its one-step purification using HPLC, and the morphology characterization of the purified peptide. 

Recently, Yoo et al. published a protocol where the Aβ(M1-42) peptide was purified using reverse-phase high-performance liquid chromatography (HPLC) [14]. We have further expanded and optimized this protocol with the following changes to provide improved versatility to the method: 1) We have expressed the pET- Sac-Abeta(M1-42) plasmid in Rosetta(DE3)pLysS cells (a BL21 derivative designed to enhance the expression of eukaryotic proteins) in addition to the BL21(DE3)pLysS cells; 2) We have performed the purification of the peptide with HPLC using the commonly available C18 columns with optimized solvent system conditions; 3) We have provided characteristic details of the peptide with Matrix-Assisted Laser Desorption/Ionization Time-Of-Flight Mass Spectrometry (MALDI-TOF MS) and verified its characteristic aggregate formations in different conditions by western blotting and Atomic Force Microscopy (AFM) to warrant future biological use. 

## 2. Experimental Design

The protocol detailed in this paper can be divided into five parts (Figure 1). The pET-Sac-Abeta(M1-42) plasmid is isolated (Part 1), transformed in the competent *E. coli* (Part 2), and the expression of Aβ(M1-42) peptide is induced in a large quantity of bacterial culture (Part 3). Expressing human Aβ(M1-42) in *E. coli* is a highly efficient and feasible method to produce large quantities of the peptide. Our protocol can be completed within a week’s time (~ five days if starting from the beginning or three days if using the transformed bacteria for generating more cultures for peptide purification) and a yield of around 15–20 mg of the peptide per 1 L of the culture can be obtained. The equipment utilized for the expression and purification of the peptide are readily available in not only biological labs, but also in synthetic chemistry and chemical biology labs, therefore promoting widespread use of the method.

The expressed peptides are present as inclusion bodies within the bacterial cells that are then isolated by repeated sonification and lysed with Tris/EDTA buffers followed by dissolution of the peptide into 8M urea buffer (Part 4). The peptide is ultimately purified by reverse-phase HPLC using the C18 columns with an optimized solvent system for peptide purification (Part 5). C18 columns are popular and conventionally used for the separation of small molecules and low molecular weight peptides. Typically a relatively large (43 residues) and hydrophobic peptide like Aβ(M1-42) [3] is not separated with a C18 column. However, heating the column between 60–80 °C results in a single peak during separation of pure Aβ(M1-42) peptide with yields similar to alternative protocols [12,13,14]. The lyophilized peptide is characterized by mass spectrometry to verify the identity of the peptide and to detect any impurities present. The aggregation characteristics of the recombinant Aβ(M1-42) are similar to the synthetic Aβ42 peptide as per AFM and western blotting with monoclonal antibody.

### 2.1. Materials


pET-Sac-Abeta(M1-42) plasmid (Addgene, Watertown, MA, USA; Cat. no.: 71875)Tryptone (Fisher Scientific, Waltham, MA, USA; Cat. no.: BP1421-500)Yeast Extract (Fisher Scientific, Waltham, MA, USA; Cat. no.: BP9727-500)LB Agar (Miller, Granulated) (Fisher Scientific, Waltham, MA, USA; Cat. no.: BP9724-500)Ampicillin (VWR International, Radnor, PA, USA, Cat. no.: 80055-786)Chloramphenicol (Fisher Scientific, Waltham, MA, USA; Cat. no.: AAJ67273AB)Plasmid Miniprep System (Promega, Madison, WI, USA; Cat. no.: A12222)Rosetta™(DE3)pLysS Competent Cells (MilliporeSigma, Burlington, MA, USA; Cat. no.: 70956-3)Isopropyl β-D-1- thiogalactopyranoside (IPTG) (Fisher Scientific, Waltham, MA, USA; Cat. no.: 15-529-019)Tris hydrochloride (Tris-HCl) (Fisher Scientific, Waltham, MA, USA; Cat. no.: BP153-500)Urea (Invitrogen, Carlsbad, CA, USA; Cat. no.: 15505-035)0.22 μm non-sterile hydrophilic PVDF syringe filter (Fisher Scientific, Waltham, MA, USA; Cat. no.: 09-719-000)HPLC grade Acetonitrile (Fisher Scientific, Waltham, MA, USA; Cat. no.: A998-4)Purified anti-β-Amyloid, 1-16 Antibody (BioLegend, San Diego, CA, USA; Cat. no.: 803001)HRP Goat anti-mouse IgG (minimal x-reactivity) Antibody (BioLegend, San Diego, CA, USA; Cat. no.: 405306)SuperSignal West Pico Chemiluminescent Substrate (Fisher Scientific, Waltham, MA, USA; Cat. no.: 34080)1,1,1,3,3,3-Hexafluoroisopropyl alcohol (Chem-Impex International, Wood Dale, IL, USA; Cat. no.: 00080)Hamilton syringe with a Teflon plunger and a sharp needle [Hamilton Company, Reno, NV, USA; Part. no.: 81343]Mica sheet (Ted Pella, Redding, CA, USA; Cat. no.: 50)Aluminum coated silicon probes with resonant frequency ~300 kHz and 40 N/m force constant (Ted Pella, Redding, CA, USA; Cat. no.: TAP300AL-G-10)Phosphate-buffered saline (PBS), 10x at pH 7.4 (Alfa Aesar, Haverhill, MA, USA; Cat. no.: J62036-K7)


### 2.2. Equipment


Benchtop incubator shaker (New Brunswick™ Excella® E24) (Eppendorf, Hamburg, Germany; Cat. no.: M1352-0000)Sonicator Ultrasonic Homogenizer (125W) with 1/4" Probe (Qsonica, Newton, CT, USA; Cat. no.: Q700-110 and 4435)CO_2_ Incubator (New Brunswick™ Galaxy® 48S) (Eppendorf, Hamburg, Germany; Cat. no.: CO48S-120-0000)Centrifuge (Sorvall LYNX 6000) with a Swinging-Bucket Rotor (BIOFlex™ HC) (Thermo Fisher Scientific, Waltham, MA USA; Cat. no.: 75006591 and 75003000)UV-Visible Spectrophotometer (JASCO, Easton, MD, USA; Cat. no.: V-730)Microplate reader for nucleic acid quantification (Take3™ Micro-Volume plate) with Gen5 Software (BioTek, Winooski, VT, USA; Cat. no.: TAKE3)Combiflash EZ prep UV/ELSD (Teledyne ISCO, Lincoln, NE, USA; Cat. no.: 218J00936)RediSep Prep 10 × 250 mm C18 100A, 5 µm column (Teledyne ISCO, Lincoln, NE, USA; Cat. no.: 692203809)RediSep Prep Guard 20 × 30 mm, C18Aq, 100A, 5 µm (Teledyne ISCO, Lincoln, NE, USA; Cat. no.: 692203805)GenPure UV/UF × CAD plus Ultrapure Water Purification System (Thermo Fisher Scientific, Waltham, MA USA; Cat. no.: 41956240)Rotary evaporator and water bath (EYELA, Keyland Court Bohemia, NY, USA; Cat. no.: N-1110 and SB-1200)Dry Bath with heating block (Thermo Fisher Scientific, Waltham, MA USA; Cat. no.: 88870002)Labconco Freezone 12 Liter Console Freeze Dry System (Lyophilizer) (Labconco, Kansas City, MO, USA; Cat. no.: 710612000)Voyager-DE PRO (MALDI-TOF mass spectrometer) (Applied Biosystems, Foster City, CA, USA;)For AFM: Veeco Multimode instrument with NanoScope V controller


## 3. Procedure

### 3.1. Preparation of Solutions

Use ultrapure water to prepare all the solutions in the protocol. Make liquid Luria Broth (LB) by dissolving 10 g tryptone, 10 g NaCl, and 5 g yeast extract in 1 L water. Autoclave the liquid LB and bring to room temperature before adding the antibiotics. Prepare solid LB as 3.2 g of LB agar mix in 100 mL water. Autoclave the broth and allowed to cool before adding the antibiotics as follows: (i) 100 mg/L ampicillin for antibiotic selection during amplification of the pET-Sac-Aβ(M1-42) plasmid in LB media and (ii) 100 mg/L ampicillin and 34 mg/L chloramphenicol for maintenance and expression of the Aβ(M1-42) peptide in Rosetta(DE3)pLysS and BL21(DE3)pLysS strains (i.e., for transformed bacteria). For solid LB preparation, pour the media with the antibiotics on to the Petri dishes and keep in the biosafety cabinet with lids open to allow the media to solidify. Prepare these plates in bulk and refrigerate at 4 °C for future use to save time during the protocol. For cell lysis steps, make two buffers: (i) Buffer A containing 10 mM Tris/HCl and 1 mM EDTA in water (pH 8.0), (ii) Buffer B containing 8 M urea, 10 mM Tris/HCl, and 1 mM EDTA in water (pH 8.0). Prepare fresh buffers each time. For peptide purification with preparative HPLC, make: (i) Solvent A as 0.1% trifluoroacetic acid (TFA) in water and (ii) Solvent B as 0.1% TFA in acetonitrile (Table 1). TFA is a common ion-pairing agent used in reverse phase-HPLC to enhance the separation of large hydrophobic peptides and is volatile and easily removed from the purified peptide. Acetonitrile is a common organic solvent used to elute peptides as it has low viscosity and is easy to evaporate off. 

### 3.2. Expression of Aβ(M1-42) Peptide. Time for Completion: 55:30 h

#### 3.2.1. Isolation of the pET-Sac-Aβ(M1-42) Plasmid. Time for Completion: 31:00 h

**Note:** The bacterial growth on solid agar as well as in liquid culture are performed at 37 °C temperature which is the optimal growth temperature for *E. coli* cells.
Streak the bacteria onto a solid LB agar plate containing 100 mg/L ampicillin using a sterile loop.**Note:** The pET-Sac-Aβ(M1-42) plasmid arrives as a bacterial agar stab culture.Keep the plate overnight for ~ 16 h in the incubator at 37 °C for the colonies to grow.The next day, pick a single colony from the plate (Figure 2A) using a sterile loop or a sterile 10 μL pipet tip and inoculate into 5 mL of liquid LB containing 100 mg/L ampicillin.Keep the culture in a shaking incubator at 220 rpm and 37 °C overnight for ~ 16 h.The next day, isolate the plasmid (Figure 1, Part 1) from the culture using the plasmid isolation miniprep kit following instructions per the user manual.Measure the concentration of the plasmid at 260 nm absorbance using a spectrophotometer for nucleic acid quantification. **Note:** Typical plasmid yield is around 45-55 ng/µL from 5 mL liquid LB culture.

#### 3.2.2. Transformation of pET-Sac-Aβ(M1-42) Plasmid into Competent E. Coli by Heat Shock Method. Time for Completion: 01:30 h


1.To transform the Aβ(M1-42) plasmid into *E. coli* (Figure 1, Part 2), thaw the frozen vials of Rosetta(DE3)pLysS cells or BL21(DE3)pLysS competent cells (20–50 μL) on ice.2.Once thawed, add around 1–2 μL of the isolated plasmid (to have 50–100 ng total DNA) to the cells and gently flick the tube a few times to mix the plasmid with the cells.3.Incubate the cells and the plasmid mixture on ice for 20 min and then place the tube in a 42 °C water bath for 45 s to facilitate the transformation of plasmid into the cells via heat shock method.4.After heat shock, immediately place the tubes on ice for 2 mins.Inoculate the transformed bacteria into 500 μL of liquid LB media without any antibiotics and keep the tube in the 37 °C shaker for 1 h at 220 rpm.
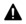
 **CRITICAL STEP:** The transformed bacteria is first inoculated in liquid LB media without any antibiotics for 1 h to allow the bacteria to express the antibiotic resistance proteins necessary for future steps.


#### 3.2.3. Expression of Aβ(M1-42) Peptide by the Transformed *E. Coli.* Time for Completion: 23:00 h


1.After 1 h, spread around 25–30 μL (out of 500 μL) of the transformed cells onto solid LB agar plates containing 100 mg/L ampicillin and 34 mg/L chloramphenicol using a sterile glass spreader.2.Let the plates sit in the biosafety cabinet for 5 min to allow the cells to absorb on the solid LB.3.Next, keep the plates overnight in the incubator at 37 °C for the transformed colonies to grow.
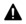
 **CRITICAL STEP:** It is best to use a lower volume of the transformed cells (25–30 μL) as this results in more single colonies that are easier to pick and prevents overcrowding on the plate.4.The next day, pick a single colony from the plate using a sterile loop or a sterile 10 μL pipet tip and inoculate into 5 mL of liquid LB containing 100 mg/L ampicillin and 34 mg/L chloramphenicol (first culture).5.**OPTIONAL STEP:** At the same time, pick another single colony of the transformed bacteria and inoculate into a second 5 mL liquid LB media containing 100 mg/L ampicillin and 34 mg/L chloramphenicol for overnight growth (second culture). This culture will be used to make frozen glycerol stocks for future use.6.Place both the cultures in the shaking incubator at 220 rpm at 37 °C overnight (16 h) (Figure 2B).7.The following day, inoculate the first 5 mL culture into a large 2 L Erlenmeyer flask containing 1 L liquid LB media with 100 mg/L ampicillin and 34 mg/L chloramphenicol (Figure 2C).8.Keep this 1 L culture at 220 rpm and 37 °C until the cell density reaches an optical density (OD) value between 0.40 to 0.45 at 600 nm (OD_600_).
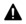
 **CRITICAL STEP:** The BL21(DE3)pLysS cells reach an OD_600_ between 0.40 to 0.45 in 3 h time, whereas the Rosetta(DE3)pLysS cells requires around 3.5–4.0 h to reach an OD_600_ between 0.40 to 0.45. It is best to measure the OD of the culture at regular 20 min intervals starting from the 3-h time point before proceeding to the next step.9.**OPTIONAL STEP:** The second 5 mL culture of transformed bacteria is used to make 25% glycerol stocks by adding 500 μL of 50% glycerol to 500 μL bacterial culture and is frozen at −80 °C for future use. A −80 °C frozen glycerol stock of the transformed bacteria is thawed for use in the future for inoculating 5 mL liquid LB containing 100 mg/L ampicillin and 34 mg/L chloramphenicol to grow additional cultures of the transformed bacteria containing the Aβ(M1-42) plasmid. **Note:** Frozen aliquots of 250–500 µL glycerol stocks may be stored for future use. It is recommended that the stored aliquots of glycerol stock be thawed 1–2 times only. Additionally, some of the frozen stock may be scrapped and thawed for inoculation.
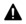
 **CRITICAL STEP:** This step serves as a starting point for all future experiments performed for the isolation of the Aβ(M1-42) peptide (Figure 1, Part 3). It is important to note that this step is critical for reducing the time taken for the entire protocol along with saving the reagents used for peptide expression.10.Once the OD_600_ of the 1 L culture reaches 0.40 to 0.45, induce protein expression by adding isopropyl β-D-1- thiogalactopyranoside (IPTG) to obtain a final concentration of 0.1 mM in 1 L of the liquid LB media. **Note:** Add 1 mL of 0.1 M IPTG stock solution prepared in water to obtain a final concentration of 0.1 mM in 1 L LB media.**Note:** IPTG induction must occur early in the exponential growth phase to form insoluble inclusion bodies, which are essential for the isolation procedure.11.Keep the culture again on the shaking incubator at 220 rpm and 37 °C for additional an 4 h in the presence of IPTG to allow the cells to express the Aβ(M1-42) peptide.12.After 4 h, centrifuge the 1 L culture at 7068× g at 4 °C for 25 min.**Note:** The cultures are centrifuged at 4 °C temperature to arrest cell growth and metabolism.**Note:** Our centrifuge allowed for a maximum speed of 7068× g with the corresponding swinging bucket rotor (see Section 2.2. Equipment). Hence, this speed was used to collect the transformed cells from the 1 L cultures. The cells may also be pelleted by centrifuging the cultures at 2800× g if using a JA-10 rotor [14].13.Discard the supernatant liquid LB and resuspend the pelleted cells in 25 mL of 1× PBS and transfer the thick cell suspension to a 50 mL falcon tube using a 10 mL pipet.14.Centrifuge the cells at 7068× g at 4 °C for 25 min and discard the 1× PBS supernatant.15.
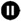
**PAUSE STEP:** Store the pelleted cells at −80 °C until the next day or when ready for cell lysis.


### 3.3. Aβ(M1-42) Peptide Purification Using Reverse-Phase HPLC. Time for Completion: 05:40 h

#### 3.3.1. Cell Lysis and Resuspension. Time for Completion: 02:24 h

**Note:** Cell lysis steps involving sonication and centrifugation is performed at 4 °C to prevent denaturation of the protein. It is critical for the pH of the buffers to be set at 8.0 to improve solubility of the peptide and prevent its aggregation. The peptide is thus maintained in alkaline conditions since it is known to aggregate at lower pH of 5.5. with lower solubility [15].
1.To lyse the cells (Figure 1, Part 4), resuspend the cell pellet in 25 mL Buffer A. Cut the tip off a 1 mL pipette tip to efficiently dissociate the thick pellet in the buffer.2.Disrupt the cell pellet mechanically by mixing the cells with Buffer A. Place the tube in an ice bucket containing ice and water and introduce the sonicator probe into the cell mixture. **Note:** Ensure that the cell mixture remains cold throughout the sonication.3.Sonicate the cells at 30 s pulse with an amplitude of 60% for 2 min until the lysate appears homogenous. **Note:** Four 30 s on/off cycles for a total of 4 min.
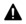
 **CRITICAL STEP:** To increase the lysis efficiency, sonicate the cell pellet in the original 50 mL frozen falcon tube since transferring the cell mixture to containers with a large surface area reduces the lysis efficiency.4.Centrifuge the sonicated mixture at 7068× g for 25 min at 4 °C and discard the supernatant.**Note:** The sonicated mixture was centrifuged at 7068× g based on the maximum speed allowed on our centrifuge and swinging bucket rotor (see Section 2.2 Equipment). This speed was sufficient to collect the pellet from the cell lysis at the bottom of the tube. The sonicated mixture may also be centrifuged at higher speeds of up to 38,000× g if using the JA-18 rotor to collect the pellet [14].5.Repeat the sonication and centrifugation steps (steps 1 to 4) three more times.6.Resuspend the pellet in 20 mL of freshly prepared Buffer B and sonicate as above until the solution appears clear.
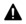
 **CRITICAL STEP:** Purification should be performed immediately after the peptide is dissolved in 8 M urea solution (Buffer B). Extended exposure of the peptide to urea is known to cause carbamylation of lysine residues.
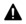
 **CRITICAL STEP:** Due to the inconsistencies of mechanical lysis using the sonicator, the solution may appear to be cloudy. **Note:** Prepare fresh Buffer B every time. The pH of the buffers is critical for the complete dissolution of the peptide fraction.7.Centrifuge the solution at 7068× g for 25 min at 4 °C to remove any foam that may have appeared during the sonication step and to pellet unwanted insoluble cell debris, if any. Finally, filter the supernatant through a 0.22 μm non-sterile hydrophilic PVDF syringe filter using a 30 mL syringe to obtain a clear solution. 8.**OPTIONAL STEP:** Prior to purification, MALDI-TOF MS may be performed on the urea-solubilized recombinant peptide to confirm the presence of Aβ(M1-42) in the solution (data not shown). Since the solution contains additional salts, the solution must be passed through a C18 Zip Tip resin to remove the salts in order to obtain a clear spectrum.

#### 3.3.2. Peptide Purification Using Reverse-Phase HPLC. Time for Completion: 03:16 h

**Equipment set-up for peptide purification**: Heat a water bath containing reverse osmosis water using a commercially-available sous vide between 60 to 80 °C (Figure 3A). Completely submerge both the guard and the primary columns in the water bath (Figure 3B). It is recommended to set up the water bath with the submerged columns at least 4 hours prior to purification to allow the water bath to reach 60 to 80 °C and the temperature of the columns to equilibrate. **Note:** As it is common for many silica-based C18 columns to degrade at higher temperatures, different columns have different recommended heating limits. Refer to your column manual or manufacturing guide to identify the column’s recommended heating limit. Heating of the guard and primary columns is necessary to prevent the recombinant peptide from sticking to the column and to improve yield of the peptide. 

Place the solvent lines from the CombiFlash HPLC instrument into the Solvent A and Solvent B bottles. Next, prime the system with Solvent B followed by Solvent A to clear the solvent lines of any previous residual solvents. Then clean the column by injecting 4 mL of Buffer B into the 5 mL injection loop (Figure 3C) and run the solvent gradient for the cleaning protocol provided in Table 1.
Place a rack of clean and dry 18 × 150 mm glass test tubes (Figure 3D) in the instrument.Run the solvent gradient as per the cleaning protocol in Table 1 at a flow rate of 5 mL/min.Equilibrate the column with the starting solvent system in Table 2 at a flow rate of 5 mL/min. And inject 4 mL of the filtered solution obtained from Section 3.2.1 step 8 into the HPLC injection loop for separation (Figure 3C).Run the solvent gradient described in Table 2 and collect peaks detected at 214 nm. **Note:** Aβ(M1-42) typically elutes at 26 min.Upon completion of the purification protocol, clean the column with the solvent gradient described in Table 1.Repeat the cleaning and purification steps three more times or until all the solution is used.Combine the collected Aβ(M1-42) fractions eluted at 26 min (Figure 4).Evaporate off the acetonitrile under reduced pressure at 65 °C using a rotary evaporator until a cloudy aqueous solution remains.Freeze the solution at −80 °C and then submerge in liquid nitrogen for 5 min.Perform overnight lyophilization at −90 °C at 0.003 mbar pressure to obtain the white Aβ(M1-42) powder (Figure 4).
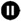
**PAUSE STEP** Store the lyophilized peptide at −80 °C until further characterization.

### 3.4. Characterization of Aβ(M1-42)

#### 3.4.1. Characterization of Aβ(M1-42) Using MALDI-TOF MS and High-Resolution LC-MS. Time for Completion: 00:45 h


Dissolve a small quantity of Aβ(M1-42) in water and dilute in water until the sample is approximately 100 μg/mL.Thoroughly mix 1 μL of the Aβ(M1-42) solution (analyte) with 1 μL of α-cyano-4-hydroxycinnamic acid (CHCA) matrix solution (10 mg/mL CHCA in 0.1% TFA).Spot the analyte/matrix mixture on a MALDI target plate and allow to dry.Obtain the MALDI-TOF mass spectra on a Voyager-DE PRO from 1000–22000 Da in the positive ion mode with an accelerating voltage of 25,000 V (Figure 5).Further dilute the analyte solution prepared for the MALDI-TOF MS and inject into the Agilent 6550 iFunnel Q-TOF LC-MS in positive mode and fragment using electrospray ionization (ESI) with a fragmentor voltage of 175V (Figure 6).


#### 3.4.2. Western Blot Characterization of the Purified Aβ(M1-42) Peptide. Time for Completion: Two Days


To perform western blotting, the lyophilized peptide from HPLC was dissolved in 1,1,1,3,3,3-hexafluoro-2-propanol (HFIP) to prepare monomers of Aβ(M1-42) as per the previously established protocols [16,17].When the peptide is dried overnight in the chemical hood, dissolve 1 mg of the peptide in 221 μL DMSO to obtain a final concentration of 1.0 mM.Load the peptide on 12% Sodium Dodecyl Sulfate Polyacrylamide Gel Electrophoresis (SDS-PAGE) at different concentrations (10, 20, and 40 μg per 40 µL total well volume) and run the gel at 115 V for 1 h and 20 min or until the loading buffer line reaches the bottom of the gel.Transfer the gel to a nitrocellulose membrane at 10 V for 35 min.After transfer, boil the membrane in PBS for 5 min, then incubate on a rocker with blocking buffer (5% milk in Tris-buffered saline, 0.1% Tween 20 (TBST)) for 1 h.After blocking, incubate the membrane in the blocking buffer containing the 6E10 monoclonal antibody (with target specificity to the human Aβ peptide) overnight on a rocker at 4 °C.The next morning, wash the membrane 3 times for 10 min each with TBST and incubate with the secondary antibody (HRP-conjugated goat anti-mouse antibody) on the rocker for 1 h.Washed again three times for 10 min each with TBST and develop in the dark room using chemiluminescence reagents as per the manufacturer’s protocol.Run the synthetic Aβ(1-42) in the same gel with the same protocol for comparison (Figure 7).


#### 3.4.3. Characterization of Aβ(M1-42) Oligomers by Atomic Force Microscopy. Time for Completion: Two days

**Preparing Aβ(M1-42) monomers**: The synthetic Aβ(1-42) and the lyophilized Aβ(M1-42) powder are used to prepare 1 mM solution with 215 μL of 1,1,1,3,3,3-Hexafluoro-2-propanol (HFIP). Handle the HFIP carefully using a 1 mL glass Hamilton syringe with a Teflon plunger and a sharp needle. Start by incubating the clear Aβ(M1-42) and synthetic Aβ(1-42) solution at room temperature for 30 min. Next, transfer 100 μL aliquots (~0.45 mg) to microcentrifuge tubes and leave the tubes open in the fume hood overnight for the HFIP to evaporate. The next morning, dry the sample under high vacuum for 1 h without heating to remove any remaining traces of HFIP and moisture. This results in thin clear films of monomeric peptides at the bottom of the tubes which are stored at −80 °C until further use.
1.To evaluate the aggregation property of the recombinant Aβ(M1-42) with respect to the synthetic Aβ(1-42), the protocol established by Stine et al. is used to prepare Aβ oligomers and fibrils [16,17].2.In brief, allow the tubes containing monomeric peptide films to equilibrate at room temperature for a few minutes and prepare 5 mM of synthetic Aβ(1-42) and recombinant Aβ(M1-42) DMSO stocks by adding 20 μL of cell-grade DMSO to each tube containing ~0.45 mg of the peptide.3.Pipet the solution thoroughly by scraping down the sides of the tube and vortexed for ~30 s followed by bath sonication for 10 min to ensure complete resuspension of the peptide film.
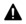
 **CRITICAL STEP:** This stock solution is used immediately as the starting material for oligomeric Aβ preparation.4.To prepare Aβ oligomers, mix 2 μL of the freshly resuspended 5 mM of synthetic Aβ(1-42) and recombinant Aβ(M1-42) in DMSO with 98 μL of 1× PBS (filtered) to make 100 μM solution. To prepare Aβ fibrils, mix 98 μL of 10 mM HCl (filtered) to 2 μL of freshly resuspended 5 mM of recombinant Aβ(M1-42) in DMSO.5.Vortex the solution thoroughly for 15 s and incubate at 4 °C (for oligomers) and at 37 °C (for fibrils) for 24 h.6.After 24 h, prepare the samples for AFM with proper sterile technique in the hood as follows: Dilute the 100 μM samples to a concentration of 30 μM in filtered water.7.Mount the mica sheet on 15 mm stainless steel pucks.8.Immediately before sample plating, remove a few layers of the mica sheet using adhesive tape to reveal a featureless surface for the absorption of the peptide.9.Next, pretreat the mica surface with ~5–8 μL of filtered 1M HCl for 30 s and rinse with 2–3 drops of ultrapure water (filtered) using the 1 mL syringe.10.
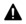
**CRITICAL STEP:** Hold the mica plate at a 45° angle to wash with drops of water. Immediately after cleaning, spot the peptide sample onto mica and incubate for 3 min.11.Gently rinse the mica with 2–3 drops of water using the 1 mL syringe and dry with several gentle pulses of clean compressed air or nitrogen gas.12.Incubate the samples at room temperature for a few hours until imaging.13.Perform the AFM imaging using a multimode AFM equipped with aluminum-coated silicon probes with ~300 kHz resonant frequency and 40 N/m force constant under the tapping mode.14.Finally, perform image analysis using the NanoScope Analysis software.

## 4. Expected Results and Discussion

A handful of methodologies have been published in recent years that show expression of Aβ42 in *E. coli* using standard molecular biology techniques, such as transformation and bacterial cell culture. The major differences among these methodologies have been the isolation and purification methods used to obtain purified peptide samples. The purification of the peptides expressed from cells is challenging due to the many different techniques proposed combined with limited resources available in a traditional biochemistry or chemical biology laboratory. For example, previously, nickel affinity chromatography was used to purify Aβ42 fusion proteins with N-terminal His-affinity tag [18] or as an extended polypeptide of His-tagged ubiquitin [19]. Both these methods require additional steps from which Aβ42 has to be eventually cleaved requiring extra time, reagents, as well as comprising on the final peptide yield. Another method used NaOH treatment followed by ultracentrifugation to isolate and purify the insoluble inclusion bodies expressed in the *E. coli* [20]. This purification method reduces the peptide purity and yield (~ 4 mg of peptide obtained). Walsh et al. [12] used ion exchange chromatography and Yoo et al. [14] used preparative HPLC equipped with a C8 column for Aβ(M1-42) purification. 

In this paper, we present a highly detailed alternate and refined approach for the rapid, easy, and low-cost production and purification of recombinant human Aβ42. By using alternate tools for peptide purification, we provide additional versatility to the protocol. Expected results and some critical points of consideration during the protocol are as follows. During the cell lysis and resuspension step, it is important to avoid leaving the peptide in 8M urea after cell lysis for a long period of time. Exposing the lysate to a high concentration of urea solution is known to cause carbamylation of the lysine residues [21]. Carbamylation can be seen in the MALDI-TOF MS by the presence of a secondary peak m/z 43 higher than the peak m/z 4642.50 corresponding to [M+H]^+^ (Figure 5B). Our HPLC system has a 5 mL injection loop, which limits the amount of sample that can be loaded onto the column. In Yoo et al. method, the cell lysate in 8 M urea was further diluted before injecting in the column [14]. We decided not to dilute the urea-solubilized fraction of the cell lysate in our protocol due to the smaller volume of the injection loop allowing for fewer batch runs thereby reducing the time taken for purification without reducing the purity of the peptide. The mass characterization of the final peptide shows that the final peptide obtained with very few impurities (Figure 6). 

Using western blotting of recombinant Aβ(M1-42) and synthetic Aβ42, we demonstrated that the purified peptide mixture contains large quantities of monomers as seen in the 4 kDa region (Figure 7). A higher concentration of the peptide shows oligomeric forms of the peptide that appear between 14–17 kDa (trimers and tetramers). Overall, we show that the recombinant Aβ(M1-42) peptide epitope can be recognized by the monoclonal 6E10 antibody that is specific for human Aβ42, suggesting future biological use of our recombinant Aβ(M1-42). Interestingly, the recombinant Aβ(M1-42) peptide showed higher molecular weight (HMW) oligomeric bands in the 38–49 kDa region that were not visible with synthetic Aβ42. Increased levels of such HMW oligomers are seen in the cerebrospinal fluid samples from Alzheimer’s patients [22]. Thus, HMW oligomers are of huge importance for understanding the etiology of Alzheimer’s disease [23]. We further characterized the aggregation property of recombinant Aβ(M1-42) using atomic force microscopy (AFM). The recombinant peptide formed large oligomers of different sizes (some greater than 20 nm, not shown) compared to the synthetic peptide that formed mostly uniform oligomers during the same time (Figure 8). The recombinant Aβ(M1-42) also formed characteristic fibrils similar to those formed by synthetic Aβ42 [17]. It has been previously shown that the recombinant Aβ42 aggregates faster and is more neurotoxic than synthetic Aβ42 [11]. Overall, the recombinant Aβ(M1-42) formed characteristic oligomers under similar aggregation conditions as the synthetic Aβ42. Further characterization may be necessary to evaluate the concentration-dependent aggregation kinetics of the recombinant Aβ(M1-42) [13].

Thus, the Aβ(M1-42) peptide could be used for several downstream in vitro and in vivo applications such as cell-based drug screening, neuroinflammation cell culture, animal models, etc., and for the synthesis and development of novel Aβ-related biorthogonal chemical and fluorescent probes to facilitate the advancement of neurological disease research.

## Figures and Tables

**Figure 1 mps-02-00048-f001:**
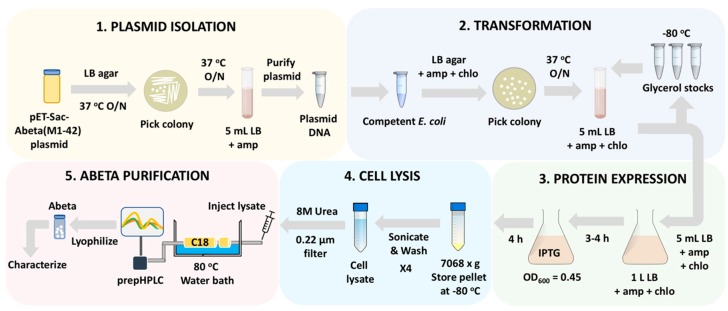
Schematic diagram illustrating the experimental protocol for the expression and isolation of recombinant human Aβ(M1-42) peptide from competent *E. coli*. The protocol can be divided into five main parts: **Part 1.** Isolation of the pET-Sac-Aβ(M1-42) plasmid from the glycerol stock; **Part 2.** Transformation of BL21(DE3)pLysS and Rosetta(DE3)pLysS competent cells with the isolated pET-Sac-Aβ(M1-42) plasmid; **Part 3.** Expression of the Aβ(M1-42) peptide in 1 L liquid LB culture; **Part 4.** Harvesting and lysis of the cells using a probe sonicator followed by resuspension of the cell lysate in 8 M urea; and **Part 5.** Purification of the Aβ(M1-42) peptide with preparative HPLC.

**Figure 2 mps-02-00048-f002:**
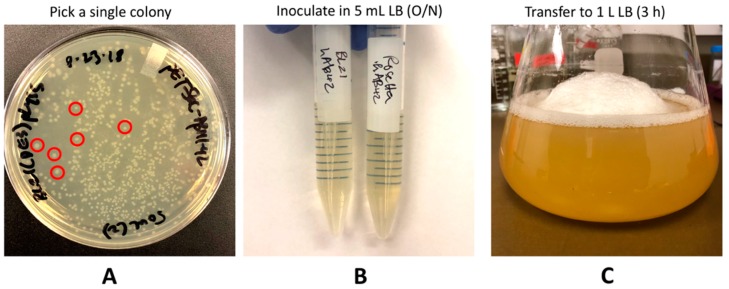
Transformed cells as colonies on the Luria Broth (LB) plate and the growing culture in liquid LB. **A.** Colonies of transformed *E. coli* on solid LB. Red circles represent single colonies. **B.** Single colony is picked from the plate and inoculated into 5 mL liquid LB and shaken at 37 °C overnight that makes a cloudy solution after incubation. **C.** The next morning, this culture is inoculated into 1 L LB to grow the cells for the next 3 to 3.5 h until the optical density (OD) reaches 0.45.

**Figure 3 mps-02-00048-f003:**
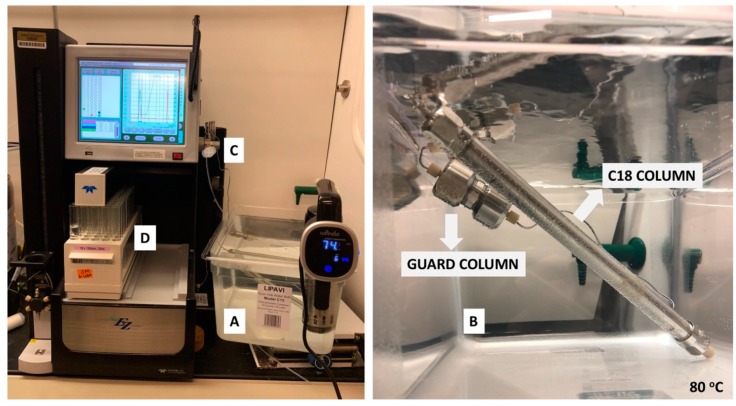
High-performance liquid chromatography (HPLC) set-up. **A.** The water bath containing filtered reverse osmosis water is heated to around 80 °C. **B.** Both columns are completely submerged in the water bath. One end of the guard column attached to the injection valve with metal tubing and the other end of the guard column attached to the inlet of the primary C18 100 Å 5 μm 10 mm × 250 mm preparative column. The primary column is fed into the Combiflash. **C.** 5 mL injection loop. **D.** Collection of Aβ(M1-42) fractions eluted at 26 min.

**Figure 4 mps-02-00048-f004:**
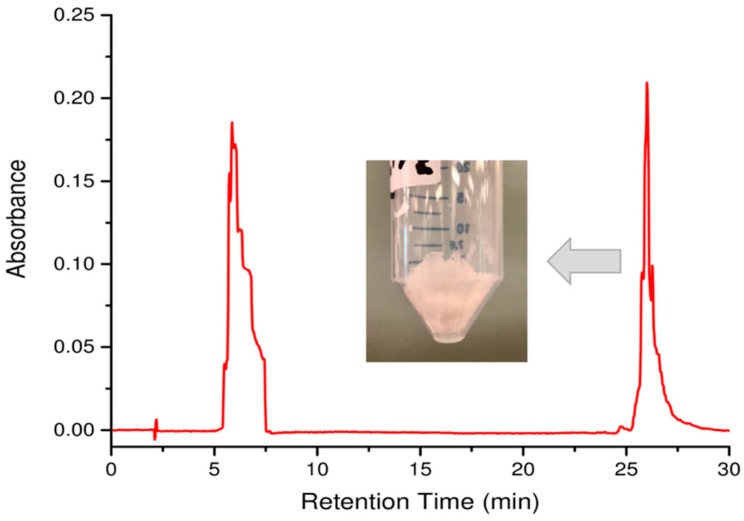
HPLC Chromatogram of urea-solubilized recombinant protein during purification. The Aβ(M1-42) peak elutes at 26 min at 95% acetonitrile with 0.1 % TFA, 5 % H_2_O with 0.1 % TFA. The urea salt from solution B elutes as a cluster of peaks between 5–7 min in 10% acetonitrile with 0.1 % TFA, 90 % H_2_O with 0.1 % TFA. The absorbance was taken at 214 nm. Inserted image of the white powder is the lyophilized peptide corresponding to the Aβ(M1-42) fractions collected at 26 min.

**Figure 5 mps-02-00048-f005:**
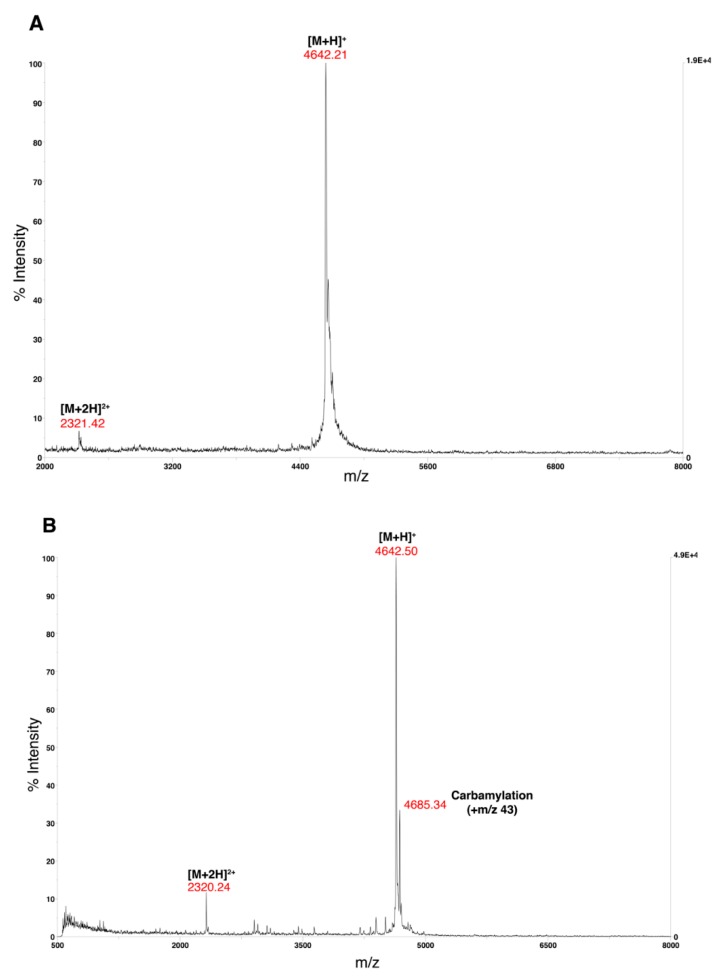
MALDI-TOF mass spectra of Aβ(M1-42). **A.** The MALDI-TOF mass spectra of lyophilized Aβ(M1-42) in the range of m/z 2000 to 8000. The Aβ(M1-42) corresponds to the m/z 4642.21 peak. **B.** MALDI-TOF mass spectra of lyophilized carbamylated Aβ(M1-42) in the range of m/z 500 to 8000. Carbamylation peak (m/z 4685.34) appears ~ m/z 43 higher than the Aβ(M1-42) at m/z 4642.50 due to the extended exposure of the peptide to urea. A Voyager De-Pro MALDI-TOF mass spectrometer in positive linear mode was used with CHCA solution as the matrix for each spectrum.

**Figure 6 mps-02-00048-f006:**
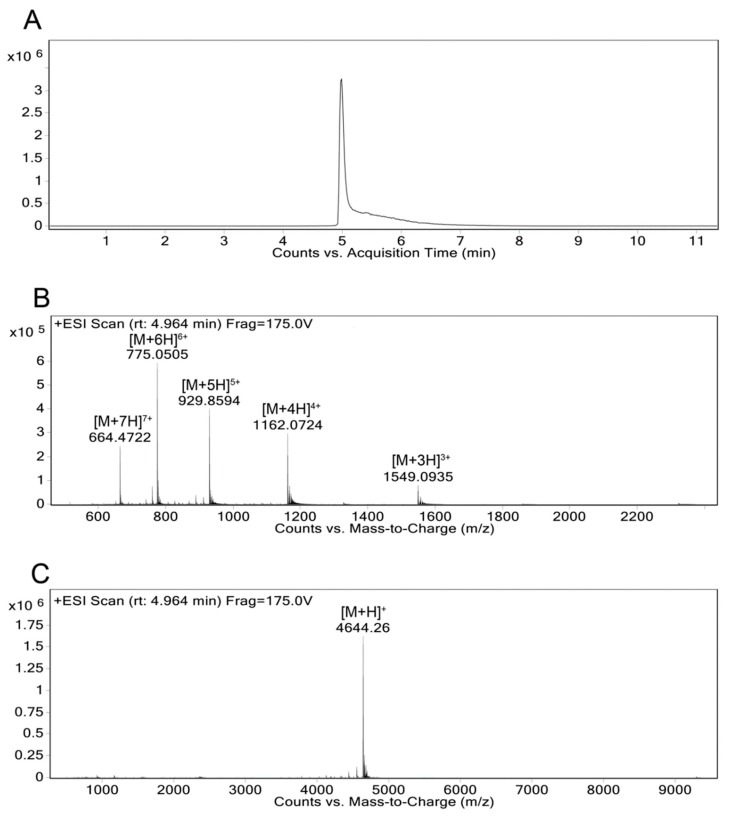
High-resolution Liquid Chromatography-Mass Spectrum (LC-MS) of Aβ(M1-42). The high-resolution mass spectrum was obtained using an Agilent 6550 iFunnel Q-TOF LC-MS in positive ion mode using electrospray ionization (ESI) with a fragmentor voltage of 175V. **A.** The chromatogram from the LC-MS showed a significant peak between 5–7 min while using 0.1% formic acid and methanol as the solvent system. **B.** Mass spectrum at time point 4.964 min resulting in the corresponding peaks: [M+3H]^3+^ (m/z 1549.0935), [M+4H]^4+^ (m/z 1162.0724), [M+5H]^5+^ (m/z 929.8594), [M+6H]^6+^ (m/z 775.0505), and [M+7H]^7+^ (m/z 664.4722). **C.** The deconvolution of the mass spectrum in panel B was performed showing the peak corresponding to [M+H]^+^ at m/z 4644.26 (calculated [M+H]^+^ of m/z was done using the PEPTIDEMASS program).

**Figure 7 mps-02-00048-f007:**
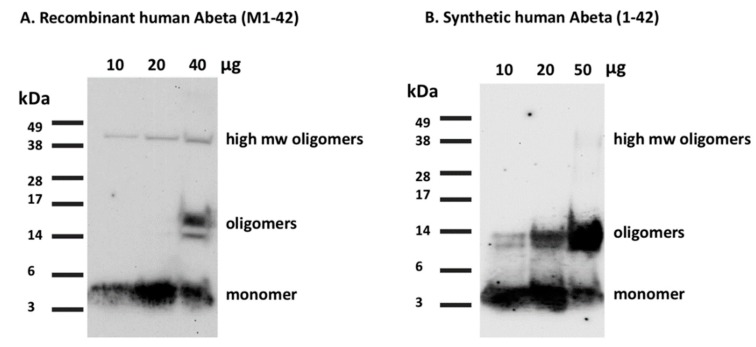
Western blot characterization of the recombinant human Aβ(M1-42) peptide. **A.** 10, 20, and 40 μg of the recombinant Aβ(M1-42) peptide was run on an SDS-PAGE gel and bands visualized using the 6E10 antibody with Western Blot. Lower concentrations of 10 and 20 μg show monomeric bands at 4 kDa while the 40 μg lane shows oligomeric bands at 14–17 kDa along with the monomeric band. All 3 concentrations show a slight amount of high molecular weight bands at 38–49 kDa suggesting a few aggregated forms of the peptide in the mixture. **B.** Synthetic human Aβ1-42 for reference shows similar monomeric bands at 10, 20, and 50 μg concentrations. More oligomers are present in the synthetic peptide.

**Figure 8 mps-02-00048-f008:**
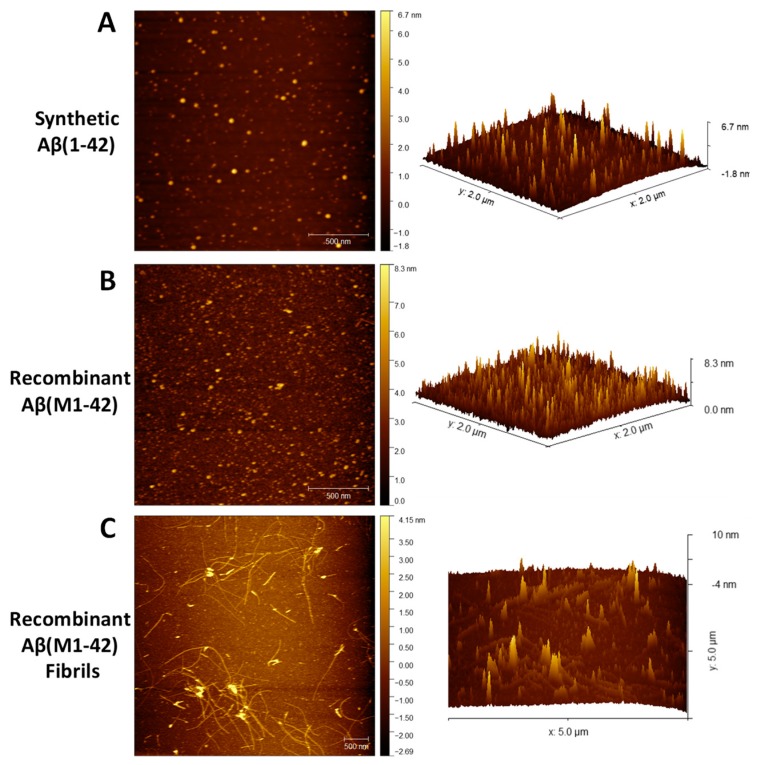
**AFM characterization of the recombinant human Aβ(M1-42) peptide.** The (**A**) synthetic Aβ(1-42) oligomers and (**B**) recombinant Aβ(M1-42) oligomers and/or (**C**) fibrils were prepared from HFIP- treated peptide films in 1× PBS buffer (pH 7.4) and analyzed by atomic force microscopy (AFM). Images (**A**) and (**B**) are 2 × 2 μm x–y scale and image (**C**) is 5 × 5 μm x–y scale. Left. 2-D image. Right. 3-D image.

**Table 1 mps-02-00048-t001:** Solvent gradient for the cleaning protocol done before and after peptide purification.

% Solvent A ^1^	% Solvent B ^2^	Elapsed Time (min)
90	10	0
90	10	5
10	90	10
10	90	20

^1^ Solvent A = H2O with 0.1% TFA ^2^ Solvent B = Acetonitrile with 0.1% TFA.

**Table 2 mps-02-00048-t002:** Solvent gradient for peptide purification.

% Solvent A ^1^	% Solvent B ^2^	Elapsed Time (min)
90	10	0
90	10	9
5	95	19
5	95	27

^1^ Solvent A = H2O with 0.1% TFA ^2^ Solvent B = Acetonitrile with 0.1% TFA.

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
