# Peer review of "Rapid, Refined, and Robust Method for Expression, Purification, and Characterization of Recombinant Human Amyloid beta 1-42"

_mps, 2019, doi:10.3390/mps2020048_

Round 1

Reviewer 1 Report

The manuscript "Rapid, Refined, and Robust Method for Expression, 3 Purification, and Characterization of Recombinant 4 Human Amyloid-beta M1-42" describes a protocol for recombinant human Aβ(M1-42) purification.

To my knowledge, no such detailed protocol has been published so far. The authors modified the protocol published in 2009. However,  since then a few new protocols were introduced. The authors should explain how their protocol differs from the protocols introduced later, for example: "Terminal Extensions Retard Aβ42 Fibril Formation but Allow Cross-Seeding and Coaggregation with Aβ42." Olga Szczepankiewicz et al., 2015.

Major issue:

The main characteristic of  Aβ(1-42)  is the formation of amyloid fibrils. To show that the protocol could be used for in vitro and in vivo assays to study Alzheimer’s disease, the authors should provide reproducible concentration-dependent aggregation kinetics and characterization of the structure and morphology of amyloid fibrils from purified  Aβ(M1-42).

Minor issue:

the use of certain parameters such as centrifugation speed (7068 g), temperature, pH time, and choice of the solvents should be better explained. 

Author Response

We would like to thank the reviewers for their insightful feedback and suggestions for our manuscript. The word file with our response is attached.

Reviewer 2 Report

This paper describes an improved method for the production of the peptide amyloid beta 1-42 (Ab42) that is a primary component of pathogenic plaques in Alzheimer’s disease. The goal of the work is to improve ease, yield and cost effectiveness of producing this peptide relative to synthetic peptide production. The method, in which the peptide is expressed in E.coli, isolated from the insoluble fraction, and further purified by HPLC, builds on work previously reported that provided the plasmid construct from which the peptide is expressed and demonstrated efficacy of E.coli expression. The major improvements in this protocol are increased expression levels due to expression in a strain modified with rare codons to improve expression of eukaryotic proteins, significant reduction in the number of purification steps, and a 4 to 5-fold increase in the yield per liter of the peptide.  The manuscript also reports the characterization of the purified peptide by analytical mass spectrometry, and analysis of its oligomerization properties by Western Blot analysis and AFM. This revealed that the purified peptide performs as expected compared to synthetic peptide, with the additional observation of a larger fraction of larger oligomers thought to have biological significance. The protocol is clearly written with detailed rationale and describes the method with sufficient detail for replication of the procedure. I have a minor comment below.

1.     Page 7, line 279-280. The term “crude lysate” is typically used to describe the mixture immediately after cell lysis, prior to separation of insoluble material by centrifugation. The material at this step has been lysed, separated by centrifugation, the soluble fraction discarded and the insoluble material resuspened in urea, then centrifuged again to remove insoluble material. Therefore, it is clearly much purified compared to crude lysate. It is more appropriate to refer to this as the urea-solubilized recombinant peptide, rather than crude lysate.

Author Response

We would like to thank the reviewers for their insightful feedback and suggestions for our manuscript. We have attached a word file with our response. 

Round 2

Reviewer 1 Report

Dear Authors, thank you for the explanation provided. I am thinking that the kinetics might strengthen the result, however, I agree that AFM image might be sufficient.

 I have no more comments to clarify. 

Author Response

Thanks for your comments. We agree that the AFM image is sufficient. The revised paper has been submitted.